# Automated detection of patients with dementia whose symptoms have been identified in primary care but have no formal diagnosis: a retrospective case–control study using electronic primary care records

Elizabeth Ford ,[1] Joanne Sheppard,[2,3] Seb Oliver,[2] Philip Rooney,[2] Sube Banerjee,[4] Jackie A Cassell [1]

► Prepublication history and supplemental material for this paper is available online. To view these files, please visit the journal online (http://dx.doi.org/10.1136/bmjopen-2020-039248).

For numbered affiliations see end of article.

**Correspondence to**
Dr Elizabeth Ford;
e.m.ford@bsms.ac.uk

## ABSTRACT

**Objectives** UK statistics suggest only two-thirds of patients with dementia get a diagnosis recorded in primary care. General practitioners (GPs) report barriers to formally diagnosing dementia, so some patients may be known by GPs to have dementia but may be missing a diagnosis in their patient record. We aimed to produce a method to identify these 'known but unlabelled' patients with dementia using data from primary care patient records.

**Design** Retrospective case–control study using routinely collected primary care patient records from Clinical Practice Research Datalink.

**Setting** UK general practice.

**Participants** English patients aged >65 years, with a coded diagnosis of dementia recorded in 2000–2012 (cases), matched 1:1 with patients with no diagnosis code for dementia (controls).

**Interventions** Eight coded and nine keyword concepts indicating symptoms, screening tests, referrals and care for dementia recorded in the 5 years before diagnosis. We trialled machine learning classifiers to discriminate between cases and controls (logistic regression, naïve Bayes, random forest).

**Primary and secondary outcomes** The outcome variable was dementia diagnosis code; the accuracy of classifiers was assessed using area under the receiver operating characteristic curve (AUC); the order of features contributing to discrimination was examined.

**Results** 93 426 patients were included; the median age was 83 years (64.8% women). Three classifiers achieved high discrimination and performed very similarly. AUCs were 0.87–0.90 with coded variables, rising to 0.90–0.94 with keywords added. Feature prioritisation was different for each classifier; commonly prioritised features were Alzheimer's prescription, dementia annual review, memory loss and dementia keywords.

**Conclusions** It is possible to detect patients with dementia who are known to GPs but unlabelled with a diagnostic code, with a high degree of accuracy in electronic primary care record data. Using keywords from clinic notes and letters improves accuracy compared with coded data alone. This approach could improve identification of dementia cases for record-keeping, service planning and delivery of good quality care.

### Strengths and limitations of this study

► Our study was representative of the UK population, using all patients in the Clinical Practice Research Datalink GOLD database who were aged >65 years and had a first recorded dementia diagnosis in 2000-2012 (N=46 713).

► However, the data used are several years old, and there may have been changes in clinical practice since.

► We trialled a number of binary classifiers, but combining machine learning approaches might have increased the accuracy achieved.

► The approach was additionally strengthened by the addition of keywords from clinic notes and text to coded information.

► A limitation is that our model was trained on cases who received a dementia diagnosis code at the end of the study period, who may differ systematically from those who never received a code.

## INTRODUCTION

Dementia is a term used to describe a group of conditions characterised by a progressive decline in cognitive function and consequent functional limitation.[1] In the UK, 1.3% of people live with dementia, the majority of whom are aged >65 years. Prevalence in all aged over 65s is 7.1%, rising from 1.8% in the 65–69 age group to more than 33% in the over 90s.[2] The challenges of dementia affect individuals, their family carers, health and care services and systems, and society as a whole, and addressing them has become

a national and international priority due to population ageing. It is expected that the cost of the disease to society will double between 2014 and 2040, reaching £55 billion per year in the UK.[2]

The UK government has recognised the benefits of an early diagnosis of dementia and has made improving rates of diagnosis a key focus of dementia policy since 2009 with the National Dementia Strategy 2009[3] and the Prime Minister's Challenge on Dementia in 2012 and 2020.[4 5] While the disease is incurable, a diagnosis gives the patient a better chance of achieving a good quality of life while living with dementia.[6] Individuals can plan and make their wishes known while they still have capacity. Services provided by healthcare and social care professionals may help people with dementia to adapt their home and identify their care needs, thus avoiding or delaying having to move into institutionalised care. Furthermore, diagnosis allows people with Alzheimer's disease to be prescribed medication and non-pharmaceutical treatments to maximise their cognitive function and quality of life. These treatments may have a particular benefit in the early stages of the illness.[7 8]

In recognition of this, general practitioners (GPs) in the UK are incentivised to initiate the diagnosis of dementia and to keep a register of patients diagnosed with dementia so that they can provide care proactively.[9] GPs are likely to be the first to recognise or be consulted about dementia symptoms, as 98% of the UK population is registered with a GP to receive primary healthcare services. If GPs suspect dementia, they refer the patient to a specialist memory assessment service where a full diagnostic screening is carried out. If the patient is diagnosed with dementia, this is communicated back to the GP who then continues the patient's care. GPs are reimbursed for ensuring a face-to-face review for each patient with dementia every 12 months. Data collected in GP patient records about dementia cases (using a prescribed set of Dementia Read codes) are used by local Clinical Commissioning Groups (CCGs) as well as national bodies such as National Health Service (NHS) England and Public Health England to monitor local and national diagnosis rates and services.[10]

Based on an epidemiological study conducted by the Medical Research Council in 1998, a prevalence of 6.6% was estimated in people aged >65 years.[11] However, based on dementia registers in the UK, the observed prevalence was found to be 3%.[11] This gap has closed substantially due to schemes like the National Dementia Strategy,[3 12] but coded diagnoses of dementia in GP patient records still only identify two-thirds of cases, compared with the expected prevalence,[10 13] and the estimated median time to diagnosis being recorded from dementia onset is 3.5 years.[14] While the diagnosis rate in the UK is higher than in many countries (eg, Lang *et al*[15] found an average rate of undetected dementia of 62% across Europe, North America and Asia), a third of people with dementia may not be receiving appropriate care.

Qualitative studies with GPs have indicated that GPs are sometimes reluctant to diagnose dementia[16–18] and

describe dementia as 'a complex condition that takes time to diagnose'.[19] GPs have reported that diagnosing dementia early was not particularly important and may in fact be harmful to some patients.[19] GPs tread carefully when initiating and communicating a diagnosis of dementia to their patients, as it is a stigmatised condition with what seems a bleak prognosis.[18 20] Some researchers have argued that there is not enough evidence for any intervention that provides a positive change in prognosis or well-being for individuals with dementia, and therefore a drive for earlier diagnosis is not justified.[21] In addition, as elderly patients often have multiple conditions, GPs report that it is difficult to integrate multiple clinical guidelines[22] and that 'dementia or memory problems (are) right at the bottom of the list'.[23] These findings suggest that patients with undiagnosed dementia could fall into two groups. First are those whose memory loss or cognitive symptoms have not been identified by the GP, possibly because they do not present to the GP very often, or they have not communicated any memory-related symptoms. Second, there is likely to be a group of patients who are known by primary care clinicians to be experiencing memory problems and may even have been screened for dementia or referred for memory assessment, but who have not, for many possible reasons, received a diagnostic code for dementia in their GP record.[16 20 24] For these patients who are 'known but unlabelled', there may be other indicators of dementia recognition and diagnosis in the patient record, which could enable them to be detected using an algorithm that combines multiple indicators of the disease. Following identification, the GP could then decide whether 'labelling' these patients with a dementia code was appropriate.

Reviews of data-driven dementia risk prediction models[25 26] show that the majority of work in the past has aimed at producing risk scores that estimate the patient's risk of developing dementia at some future date. Only a few of these risk scores have been developed on routinely collected primary care data, of which a notable example is Walters *et al*.[27] Automated early detection of emergent cases has received less attention. Few attempts have been made to produce models that detect dementia which is current but unrecognised by the GP; the only examples using UK primary care data (known to the authors) are currently Jammeh *et al*[28] and Ford *et al*.[29 30] Other studies have approached this problem manually, for example, Russell *et al*[31] aimed to improve recording in dementia registers by a hand-searching exercise, recognising the need to 'clean up dementia coding and records'. This successful exercise increased the identification rate for dementia in participating GP practices by 8.8%.

Following success in detecting unrecognised patients with dementia,[29] in this study we aimed to identify a range of elements in the primary care record which would identify the second group of 'known but unlabelled' patients with dementia. A second aim was to evaluate whether the model could be improved by the use of keyword information in clinic notes and letters found in the patient

records, rather than relying solely on information captured in clinical codes. Automatically detecting these patients may help improve the quality of data in dementia registers, enable ongoing care and support of individual patients with dementia, and give commissioning bodies and service planners more accurate prevalence estimates.

## METHODS
### Study design
A retrospective case–control study using routinely collected primary care patient records from England, reported according to RECORD (Reporting of Studies Conducted using Observational Routinely-collected health Data) guidelines.[32]

### Data source
This study used data from the UK Clinical Practice Research Datalink (CPRD). It was established in 1987 and now contains anonymised healthcare records from more than 7 million current patients and represents up to 13% of the UK population.[33 34] Patients are broadly representative of the UK general population in terms of age, sex and ethnicity. CPRD includes longitudinal observational data from GP electronic health record (EHR) systems in primary care practices, including medical diagnoses, referrals to specialists and to secondary care, testing and interventional procedures conducted in primary care, lifestyle information (eg, smoking, exercise) and drugs prescribed in primary care.[34] Data are captured using a structured hierarchical vocabulary called Read codes; these were developed by a UK GP, Dr James Read, in the 1980s.[35] They map to other nomenclatures such as International Classification of Diseases (ICD), Systematized Nomenclature of Medicine Clinical Terms (SNOMED-CT) and International Classification of Primary Care codes. Each Read code represents a term or short phrase describing a health-related concept. There are more than 200 000 different codes, which are sorted into categories (diagnoses, processes of care and medication) and subchapters.[36] Each clinical entity is represented by an alphanumeric code and a Read term which is the plain language description. CPRD curates two primary care databases: one is formed of patient records derived from a GP patient record computer software called VISION (CPRD GOLD) and one is formed from records derived from a GP patient record computer software called EMIS (CPRD Aurum).[33] This study used only CPRD GOLD data.

GPs also record clinical information in free-text clinic notes and letters. CPRD held these clinic notes at the time of study data extraction and made these available via either keyword search or release of full text for small samples, following three-stage manual deidentification of text and approval of a protocol. Since 2013, free-text notes and letters have not been extracted into CPRD from GP practices, and historic clinic text became unavailable to researchers in 2016. Due to this cessation of free-text

collection, this study used data of patients who had a first dementia diagnosis recorded up to the end of 2012 so that free text was available for the full period prior to their diagnosis. CPRD also offers data linkages with other NHS and administrative data sets, but no linked data were used in this study.

### Study population
CPRD provided the study team with longitudinal data from patients with a dementia code (cases) recorded between 2000 and 2012, as defined in a code list of dementia diagnostic codes (general dementia, vascular and Alzheimer's dementia codes) that we developed drawing on code lists in Russell et al[31] and Rait et al[37] and reported in Ford et al[29] (online supplemental appendix 1). For data extraction by the CPRD team, patients were required to be 65 years or older, and were required to have records available for at least 3 years prior to a first diagnosis code recorded between 2000 and 2012. All patients in the CPRD GOLD database meeting these criteria were selected to maximise sample size and reduce the possibility of selection bias. Control patients were randomly sampled from patients who had no dementia codes anywhere in their record (to reduce the likelihood that they just had a delayed diagnosis); who matched cases on age, sex and general practice; and who also had a minimum of 3 years of records before the diagnosis date of their matched case (known as the 'index date'). This resulted in a near-perfect 1:1 match between cases and controls; a few cases did not receive a match. No minimum duration of follow-up after index date was required as we did not want to bias the sample towards healthier individuals or exclude those who moved home or went into residential care (and thus changed GP practice) during follow-up. Thus, patients could have a date of death during the study as long as this was after the index date. Data collection on all patients was limited to November 2013 as this was the date of the final CPRD GOLD database build which included free text. This resulted in 47 858 cases and 47 663 controls.

The entire coded patient record was extracted for each patient, including all clinical codes, from the patient's earliest registration date in CPRD until the date of data extraction (November 2013), or the patient's date of death or the date they left the practice, whichever was earlier. In addition, CPRD ran a search through free-text notes and letters for keywords for all patients (described below) in the 24 months before the index date. For the keyword search, CPRD returned structured data comprising a count of the keywords in each keyword group, found in each free-text entry in the record. The study team did not have access to the free text of any patient for this study.

### Refining participants
To standardise the data set provided by CPRD and to optimise it for the planned analysis, further refinements were made to the sample. From the extracted data set, cases without matched controls and all patients lacking 5 years of data before the index date were removed along with

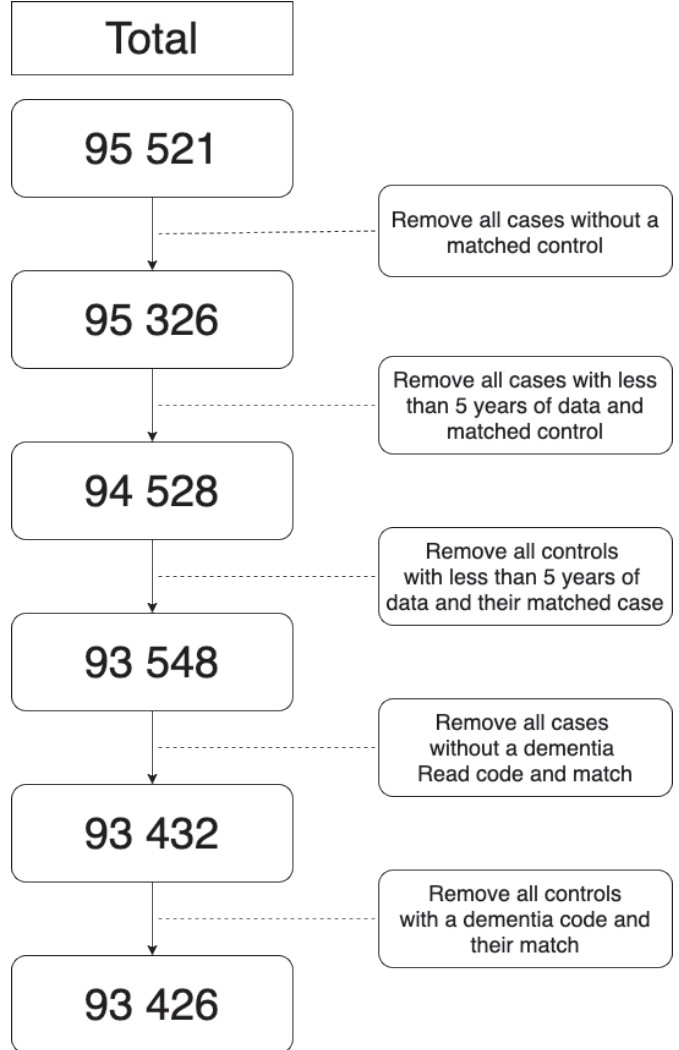

**Figure 1** Final participant selection.

their corresponding match. We removed cases for whom we could not find a dementia Read code within the study period and controls who did have a dementia code, and we removed these along with their matched patients. See figure 1 for a flowchart of final participant selection.

### Feature selection
#### Code lists
Using an expert consensus approach, with an epidemiologist, a psychiatrist and a GP, we identified a list of eight potential clinical concepts which would be indicators of the onset of dementia, initiation of the diagnostic pathway for dementia, or evidence of continuing care for dementia and which would be captured within the Read codes. Our final list of eight coded concepts entered into each model were cognitive decline; memory loss symptoms; Mini-Mental State Examination (MMSE); other cognitive screening tests; referral to memory clinic; referral to psychiatrist, neurologist or geriatrician; dementia annual review; and Alzheimer-specific prescription codes. Relevant code lists were sought from published papers, and where these were not available, lists were drawn up by

author EF and checked by author PR (online supplemental appendix 2). Although MMSE scores are sometimes recorded in structured data, we found a majority of scores were missing in our data set (56%). We decided to keep the model as simple as possible and therefore did not use MMSE scores in the analysis.

### Keywords from clinic notes and letters
We consulted with 17 GPs, asking them to suggest keywords representing terms that they might use when considering that a patient might be developing dementia. From these, we identified nine concepts: memory (suggested by 17 GPs), cognition (6 GPs), forgetful (14 GPs) MMSE or other screening test (6 GPs), confused (9 GPs), dementia (3 GPs), behaviour (5 GPs), family concern (6 GPs) and third party consultation (1 GP). We then generated a list of concept variations and spelling variations for each keyword group (online supplemental appendix 3).

### Analysis
Only data from 5 years before the index date up to the index date for each patient were used. Each patient's record was searched to find all instances of the Read codes referring to the eight features described above. If any codes were found for a feature within a patient's records, a dummy variable representing that feature was set to '1'; otherwise, they were defaulted to '0'. Similarly, if evidence of keyword was found, the dummy variable for keywords was set to '1'. A dummy variable for 'any keyword' was also included. Other studies have also favoured binary features to reduce the influence on the data of the number of times a patient visits his or her GP.[28]

Logistic regression, random forest and naïve Bayes classifiers were used as they are effective supervised machine learning methods for the analysis and classification of binary data. Logistic regression uses the logit function to find the probability of a data point belonging to a (binary) group, using one or more nominal, ordinal, interval or ratio-level independent variables.[38] The addition of the LASSO (least absolute shrinkage and selection operator) function selects the best predictors based on the residual sum of squares.[39] The random forest method is a non-parametric machine learning method for binary classification. It constructs a multitude of decision trees that best use the predictors to determine which category of output each case should be in. It holds the multitude of decision trees constant, and the output for each individual is the mode or mean prediction class of the individual trees.[40] Naïve Bayes classifiers are based on Bayes' theorem.[41] Parameter estimation for naive Bayes models uses the method of maximum likelihood, and all predictive features are treated as contributing independently of each other (ie, they are naïve to each other) to predict which category of the outcome each case belongs to.[42] These were implemented using scikit-learn with Python V.3.7.1 with 'LogisticRegression', 'BernoulliNB' and 'RandomForestClassifer' packages, with the seed set to 42 for reproducibility. The data set was split into a 70% set

for training the classifiers and a 30% testing set to assess model accuracy.

To assess the accuracy of the models, the area under the receiver operating characteristic curve (AUC) was calculated, which plots the true positive rate against the false positive rate for every possible cut-off of the predictive model. In addition, sensitivity and specificity values were calculated for model cut-offs where these values were equally weighted. Positive predictive values (PPVs) were also calculated. While the sensitivity and specificity are fixed parameters of the test, PPVs are conditional on the prevalence of the condition in the tested sample. Given the matched case–control design, our sample had a dementia prevalence of 50%, which would inflate PPVs artificially. We therefore separately calculated the PPV of the model using the estimated UK prevalence of dementia in people aged >65 years of 7.1%.[2] This was achieved using the values of sensitivity and specificity calculated above and a hypothetical sample of 100 000 patients where 7100 had dementia.

To assess which features contributed most to the models, we generated relative indices, with the most predictive feature weighting set at 1, and other features shown relative to this. This was instead of reporting the exact outputs from each machine learning algorithm (such as β weights from logistic regression) because we wanted models to be directly compared.

## Sensitivity Analysis

For the main analysis, we did not remove any controls who had indicators of dementia, as we felt this was closest to a real-world scenario. However, using only diagnosis codes and not Alzheimer's medication or dementia review codes as the case definition of dementia may inflate the performance of the model. To examine the effect of our case definition on results, we ran sensitivity analyses where control patients with any Alzheimer's medication prescriptions, or dementia annual review codes were removed from the sample, and these two code lists were removed as predictors from the model.

## RESULTS
### Study participants

The final data set consisted of 93 426 patients with a median age of 83 years (range, 65–110 years). There were 32 876 male participants (35.2%) with a median age of 81 years (range, 65–100 years) and 60 550 female participants (64.8%) with a median age of 83 years (range, 65–110 years). In all, 46 713 were cases and 46 713 were controls. The mean number of codes recorded per patient in the 5 years before the index date was 614 for controls and 586 for cases (range, 0–7281).

### Number of features by case and control group

Cases had much higher rates of all chosen features, as can be seen in table 1. The most common coded feature in dementia cases was memory loss codes (46%), followed

**Table 1** Number and proportion of cases and controls who had each feature

| Feature | No. of cases with feature (%) (N=46 713) | No. of controls with feature (%) (N=46 713) |
|---|---|---|
| **Codes** | | |
| Cognitive decline codes | 3276 (7.01) | 240 (0.51) |
| Cognitive screening test | 888 (1.90) | 124 (0.27) |
| Dementia annual review | 17 372 (37.19) | 151 (0.32) |
| Memory loss codes | 21 282 (45.56) | 1387 (2.97) |
| MMSE test | 9574 (20.32) | 600 (1.28) |
| Referral to memory assessment service | 3464 (7.42) | 117 (0.25) |
| Referral to old-age psychiatrist, neurologist or geriatrician | 12 683 (27.15) | 2227 (4.77) |
| Alzheimer's medication prescription | 10 044 (21.50) | 203 (0.43) |
| **Keywords** | | |
| Memory | 27 025 (57.85) | 3815 (8.17) |
| Cognition | 8873 (18.99) | 2178 (4.66) |
| Forgetful | 14 272 (30.55) | 3397 (7.27) |
| MMSE or screening test | 11 969 (25.62) | 2189 (4.69) |
| Confused | 20 530 (43.95) | 5334 (11.42) |
| Dementia | 23 935 (51.24) | 2848 (6.10) |
| Behaviour change/problems | 5388 (11.53) | 2132 (4.56) |
| Family concerned | 37 779 (80.87) | 23 718 (50.77) |
| Third party consultation | 706 (1.51) | 1854 (3.97) |

MMSE, Mini-Mental State Examination.

by dementia annual review codes (37%). Keywords were all common throughout cases; the most frequent were family concern (81%), memory (58%) and dementia (51%). All keywords were more common in cases than in controls, except for third party consultation, and family concern was also high in controls (51%), although not as high as in the cases.

### Model accuracy

The eight coded clinical concepts were used to train the three classifiers to predict which patients' record would end with a dementia diagnosis and which patients' record would not. Trained models were then tested on the testing data set, and the accuracy of discrimination was assessed. Using only coded variables, the three models produced a high level of correct discrimination between cases and

**Table 2** Discrimination of three models between cases and controls

| | Codes only | | | | Codes and keywords | | | |
|---|---|---|---|---|---|---|---|---|
| **Classifier** | **AUC** | **Sensitivity** | **Specificity** | **PPV at 7.1% prevalence** | **AUC** | **Sensitivity** | **Specificity** | **PPV at 7.1% prevalence** |
| Random forest | 0.90 | 0.77 | 0.92 | 0.46 | 0.94 | 0.85 | 0.92 | 0.46 |
| Logistic regression | 0.90 | 0.76 | 0.93 | 0.49 | 0.94 | 0.84 | 0.93 | 0.51 |
| Naïve Bayes | 0.87 | 0.78 | 0.91 | 0.45 | 0.90 | 0.80 | 0.91 | 0.43 |

AUC, area under the receiver operating characteristic curve; PPV, positive predictive value.

controls. The three classifiers performed very similarly, with AUCs of 0.87–0.90 (table 2 and figure 2).

We then added the nine keywords to the code-only models and found that discrimination improved up to an AUC of 0.94 in the logistic regression and random forest and 0.90 in the naïve Bayes classifier (figure 3). It can be seen in table 2 that the inclusion of the keywords increased the sensitivity of the models by around 8%, without any detrimental effect on specificity.

### Feature weighting

The relative weighting of each feature in the different models, with the most predictive feature weighting set at 1, and other features shown relative to this is shown in table 3, separated by models for the coded data only and for codes and keywords together. The table shows that the three different models had substantial differences in the weighting of the different features when using only keywords. The logistic regression weighted most highly dementia annual review codes and prescriptions of Alzheimer's medication, whereas the random forest weighted memory loss codes the highest and the naïve Bayes classifier weighted referral for memory assessment as the most informative feature.

Looking at the combined codes and keywords, similar codes were most important in the three models, whereas the most important keyword was word memory (rated highest feature in the random forest model), followed by keywords for dementia. Interestingly, the logistic regression classifier prioritised third party consultation, the only feature higher in the controls than in the cases. The naïve Bayes classifier prioritised coded variables higher than all keywords.

### Sensitivity analysis

We reran the models removing controls with Alzheimer's medication prescriptions and dementia annual review codes. We also removed these two code lists as predictors in the model. The overall model accuracy, measured by AUC, was reduced for the codes-only models (AUCs: logistic regression 0.83; random forest 0.83; naïve Bayes 0.83); this reflects the high contribution these two predictors made in the main models. However, in these models, specificity increased for all codes-only models to 0.95, which also resulted in higher PPVs (logistic regression 0.61; random forest 0.61; naïve Bayes 0.60). Results for all performance parameters for models using both codes and keywords remained similar (AUCs: logistic regression 0.93; random forest 0.93; naïve Bayes 0.89). Full results of the sensitivity analysis are given in online supplemental appendix 4.

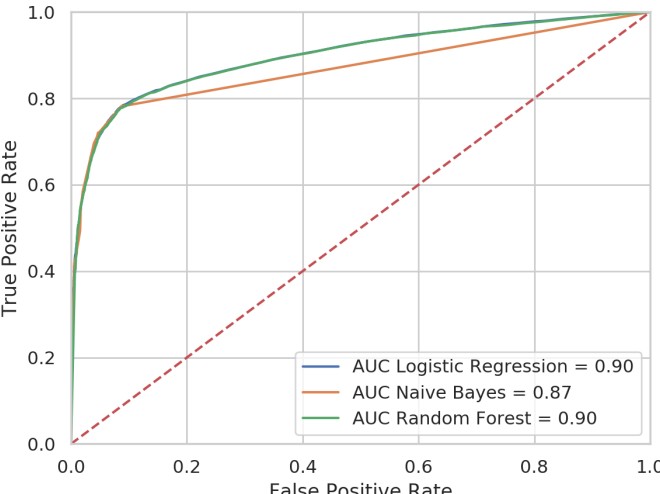

**Figure 2** ROC showing discrimination between dementia cases and controls by three machine learning models with codes only. AUC, area under the receiver operating characteristic curve; ROC, receiver operating characteristic.

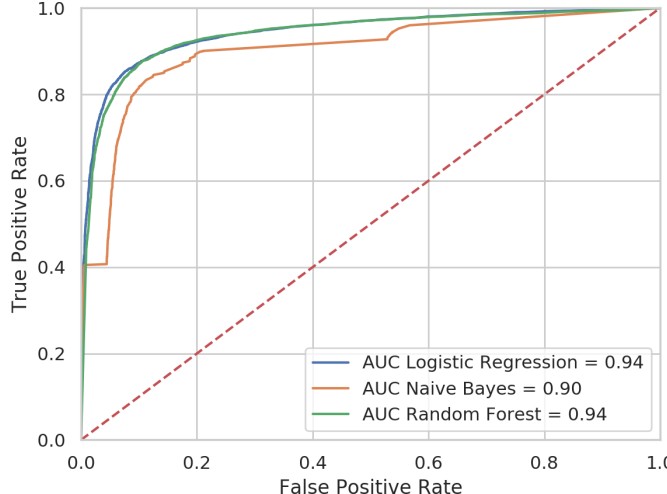

**Figure 3** ROC showing discrimination between dementia cases and controls by three machine learning models with both codes and keywords. AUC, area under the receiver operating characteristic curve; ROC, receiver operating characteristic.

**Table 3** Relative weighting of each feature in the three classifiers

| | Coded data only | | | Codes and keywords | | |
|---|---|---|---|---|---|---|
| | Logistic regression | Random forest | Naive Bayes | Logistic regression | Random forest | Naïve Bayes |
| Alzheimer's medication | 0.727 | 0.354 | 0.912 | 0.676 | 0.339 | 0.912 |
| Cognitive decline | 0.378 | 0.066 | 0.888 | 0.260 | 0.061 | 0.880 |
| Cognitive screening test | 0.025 | 0.006 | 0.997 | 0.076 | 0.010 | 0.997 |
| Dementia annual review | 1 | 0.845 | 0.957 | 0.917 | 0.888 | 0.957 |
| Memory loss codes | 0.622 | 1 | 0.593 | 0.456 | 0.964 | 0.593 |
| MMSE | 0.344 | 0.187 | 0.729 | 0.199 | 0.157 | 0.729 |
| Referral to memory assessment | 0.312 | 0.049 | 1 | 0.143 | 0.033 | 1 |
| Referral to psychiatrist, neurologist or geriatrician | 0.345 | 0.292 | 0.509 | 0.248 | 0.210 | 0.509 |
| Keyword dementia | – | – | – | 0.494 | 0.975 | 0.473 |
| Keyword memory | – | – | – | 0.251 | 1 | 0.422 |
| Keyword confusion | – | – | – | 0.238 | 0.444 | 0.363 |
| Keyword behaviour | – | – | – | 0.037 | 0.091 | 0.520 |
| Keyword cognition | – | – | – | 0.031 | 0.078 | 0.515 |
| Keyword family | – | – | – | 0.158 | 0.079 | 0.113 |
| Keyword MMSE | – | – | – | 0.019 | 0.119 | 0.515 |
| Keyword forget | – | – | – | 0.051 | 0.087 | 0.439 |
| Keyword third party consultation | – | – | – | 1 | 0.200 | 0.544 |

MMSE, Mini-Mental State Examination.

## DISCUSSION

We produced a set of binary coded variables that could be combined in a statistical model and which could accurately classify patients who went on to have a dementia diagnosis formally coded. This model could be taken forward to the clinic to identify 'known but unlabelled' patients with dementia in primary care, thus improving record-keeping, prevalence estimates and ongoing care for these patients. We also investigated whether adding in information from the clinic notes and letters to the model, in the form of a simple keyword extraction, could add further value to the approach. The addition of free-text keywords improved the discrimination of the model to an excellent AUC of 0.94, suggesting this model would perform well in the clinic. The improved performance with the addition of free-text keywords was achieved because of increased sensitivity of the model, with no loss of specificity, as has been shown in a range of other case-identification studies incorporating free text.[43] This study provides evidence that hidden data in these free-text fields can substantially improve automated detection of cases, adding weight to arguments that free-text data are valuable for research, if acceptable privacy-preserving methods for data sharing (such as automated information extraction behind a clinical firewall) can be agreed on.[44] We tried three different machine learning approaches that are suitable for binary data and found all performed similarly, although they weighted the various features

differently in their classifications. This is likely to be due to the different construction of the algorithms within each method, for example, random forest algorithms are thought to weight most highly the most common features in the data set, rather than features which are rarer but differ more substantially between the two groups. Future work could look at whether there is added value in combining the three approaches, as ensemble methods can sometimes outperform any of the single approaches alone, especially where the different methods are clearly prioritising different features to achieve their performance. However, most ensemble methods have been trialled on multiclass rather than binary classification.[45]

We also chose not to clean up our control group for evidence of dementia medication prescribing as has been done in other studies,[46] because we felt that doing so would have artificially inflated the performance of our models. However, in the literature, multiple case definitions of dementia are used in primary care EHR research, some of which incorporate Alzheimer's medication prescriptions or dementia review codes. To test the impact of our case definition choice, we ran a sensitivity analysis, removing patients from the control group who had these codes, as well as taking the code lists out of models as predictors. As these two predictors had been highly weighted in the original models, model performance in the codes-only models dropped when they were removed, although specificity increased along with PPV.

Interestingly, however, there was very little drop in performance in the models that used both codes and keywords, suggesting the keywords could make up for the loss of these two sets of codes. The case definition most useful in practice will depend on how GPs make up the inclusion criteria for their dementia registers. Where GP practices use only dementia codes for populating their case register, an algorithm which searches for Alzheimer's medication prescriptions or dementia annual review will be helpful for spotting true positive cases among their unlabelled patients with dementia.

Our study is a novel contribution to the literature. Studies have previously aimed to detect dementia that was unknown to primary care practitioners, by manual[47–49] and automated methods,[28] using data from primary care records. In addition, a recent study by Aldus et al[14] has explored rates and predictors of patients with dementia going undetected. However, to the authors' knowledge, no prediction or early detection approaches have made the distinction between patients with dementia unknown to the GP and those known to the GP but nevertheless unrecorded with a formal diagnostic code. The study by Aldus et al gives the first indication of how many of the 'undiagnosed' cohort may be 'known but unlabelled', suggesting that both unknown and known but unlabelled groups decline over time, but in that of patients with no diagnosis, around 66%–71% were not known to have symptoms, whereas 29%–34% had memory concerns or a referral noted in their patient record.[14] These figures were based on small numbers, and it would be worth investigating this further to estimate the proportions of each group. Our team has also used social science methods, interviewing and surveying GPs and patients to understand what is happening in this apparent 'diagnosis gap' for dementia in UK primary care. We suggest that this dual 'data science and social science' approach to understanding and using data in patient records for clinical decision or diagnostic support is valuable, as it exposes the underlying human decision-making and behaviour which influences how the data are created. These insights can be drawn on to improve clinical prediction models and make them more relevant for key stakeholders, which is likely to improve implementation. A further avenue for exploration with GPs, and other stakeholders such as service commissioners, is whether the results produced by this approach would be of value either for improving individual patient care or for service and resource planning, or commissioning, due to more accurate prevalence rates in patient data.

Further work should also examine the ethics and desirability of such methods in practice. Previous studies have highlighted a range of potential ethical issues that need further exploration. First, according to the principles of ethical screening by Wilson and Jungner,[50] any kind of screening programme is only ethical if there is an accepted treatment for patients with recognised disease. Weatherby and Agius argue that given no treatments that modify disease trajectory for dementia are available and

given the possibility of false positive results, population screening for dementia is ethically unjustifiable, although a stepwise process that becomes increasingly targeted might be appropriate.[51] A stepwise approach to detection may be especially important given the low PPVs found when assessing our model against a population prevalence. Ienca et al have also argued that the use of big data for early detection may lead to an inflation of the false positive risk, presumably because of algorithms being applied indiscriminately on whole populations.[52] They also draw attention to the need for clinicians or researchers to be cautious about the privacy and disclosure issues of personal data about dementia, as this would be a highly sensitive information.[52] Other authors have discussed the ethical implications of genetic testing for Alzheimer's risk in early detection or predictive models,[51 53] as well as needing to maintain the 'right not to know' for patients.[54] Drawing on these debates and other literature, Thyrian et al argue that early recognition is wanted by a majority of patients if offered in primary care, that GPs are also largely in favour[55] and that evidence-based case-finding algorithms may be ethically acceptable in primary care if tailored to both diagnostic procedures and, importantly, a postdiagnostic support package.[56]

## Strength and limitations

This study used a large sample of English patients who were all aged >65 years, and patients registered in CPRD are thought to be representative of patients in England as a whole. However, to ensure that this model is robust to different patient groups, it should now be evaluated in a new patient sample, perhaps in one which is produced from a different general practice computer system (CPRD GOLD data are harvested from VISION software). A further strength is that we specified a very simple model that dichotomised the presence of an indicator from a code list; this means the model can be readily operationalised in any GP record system and should be robust to number and order of code accrual in the patient record.

However, there are important limitations in our study design which could be assessed or addressed in future work. Notably, data collection was limited to cases identified before the end of 2012, and therefore it is unclear whether the same model would apply in current times, as the many government initiatives that have been implemented since our study's data collection may have changed clinical practice and thus recording. We plan future work to assess this and to strengthen the analysis using a cohort design and linked data from hospital records and death certificate data. We used a matched case–control design in which the contribution of age and sex to the model was controlled for in the matching and in which only one control for each dementia case was used. Using more controls per case would have increased statistical power and reduced the risk that controls were in fact undiagnosed dementia cases. We

also chose not to use MMSE score, as we aimed to create simple models with binary predictors and because the majority of MMSE scores were missing from structured data. A future cohort study where the prevalence is lower, where age can be added as a predictor, where possible false negatives can be removed from the control group and which could explore more complex predictors such as MMSE score will be a robust assessment of this model's replicability.

A further limitation is that our model was trained on cases who received a dementia diagnosis code at the end of the study period. There are likely to be a number of dementia cases who never receive a dementia code, and it remains a possibility that these patients differ in systematic ways from cases who go on to receive a code. The ability of our model to detect patients who never receive a code is therefore unknown. One possibility is that patients detected by our model will have mild cognitive impairment or another non-dementia memory complaint; this may especially be true given the low expected PPV in an unselected population of over 65s. An evaluation in a GP practice, where the predictions of the model are assessed and validated by the practising GP, would give us further information about its performance in real-life clinical settings and would help us understand more about potential false positives flagged by the model. Finally, we used a very simple keyword extraction to access the keyword data and did not incorporate any natural language processing to identify negation, hedging, subject or timing of the words we identified. We may therefore have oversampled instances of the clinical concepts being recorded about the patients; however, this would have affected both cases and controls equally and is most likely to have resulted in a bias towards the null. It was important for our study design that we chose simple feature extraction methods that could potentially be recreated in a busy clinic.

### Implications and future steps

Our model shows good accuracy in identifying patients with dementia who have been detected as likely to have dementia but who currently have no diagnostic code for the condition. As outlined above, two further validation studies are now indicated: first, a replication in a cohort study using similar primary care patient data, and second, a proof-of-concept trial in a local GP practice or primary care network. We hope that if these two further pieces of work indicate the same excellent discrimination between patients with dementia and those without, and following steps to determine the threshold resulting in high specificity and sensitivity, and identifying reasons for potential false positives, this model could be adopted by GPs and CCGs to improve the quality of their dementia registers and to aid service planning.

These findings also provide a potential way for EHR researchers to extend their dementia case identification when using primary care data for dementia research and potentially to improve research quality. Emerging evidence on the rates of undetected dementia[14] and novel methods for identifying probable but unlabelled cases such as those reported here mean that researchers could run sensitivity analyses on a range of more and less sensitive dementia definitions, rather than rely on diagnosis codes alone. We suggest exploring combinations of memory loss symptoms, Alzheimer-specific prescriptions and dementia annual review codes to extend the case definition of dementia in EHR research and to conduct validation studies on these new case identification methods and evaluate reasons for false positives. MMSE scores, where recorded, could be explored as an additional variable of particular value. As random forest and logistic regression models performed very similarly in this and other studies,[29] we would suggest either analytical approach could be valuable for classification tasks in dementia EHR research.

## CONCLUSIONS

We have shown that a simple model made up of indications of dementia symptoms, screening tests, referral and follow-up care in coded and free-text data can identify primary care patients who have recognised dementia but no diagnostic code for it in their record. This model can be taken forward for further validation and used by GPs and commissioners to improve the quality of data on patients with dementia, and as a result improve dementia care and services.

**Author affiliations**
[1]Department of Primary Care and Public Health, Brighton and Sussex Medical School, Brighton, UK
[2]Department of Physics and Astronomy, University of Sussex School of Mathematical and Physical Sciences, Brighton, UK
[3]Medical Physics and Biomedical Engineering, UCL, London, UK
[4]Faculty of Health, University of Plymouth, Plymouth, UK

**Acknowledgements** This work uses data provided by patients and collected by the NHS as part of their care and support. #datasaveslives.

**Contributors** EF, SO and JAC conceived and directed the study. EF drew up code lists and PR checked these. PR managed the data. JS conducted the analyses. PR and SO gave data analysis advice. SB and JAC gave clinical advice. EF and JS wrote the manuscript. All authors provided critical feedback on the manuscript and approved the final version.

**Funding** This project was funded by a grant from the Wellcome Trust ref. 202133/Z/16/Z.

**Competing interests** None declared.

**Patient consent for publication** Not required.

**Ethics approval** This study was approved by the Independent Scientific Advisory Committee at the Medicines and Healthcare Products Regulatory Authority, UK, protocol number 15_111_RA. Following approval, administrative permissions to access and use the electronic patient records were granted by Clinical Practice Research Datalink (CPRD.com).

**Provenance and peer review** Not commissioned; externally peer reviewed.

**Data availability statement** Data may be obtained from a third party and are not publicly available. The data that support the findings of this study are available from Clinical Practice Research Datalink (CPRD; www.cprd.com), but restrictions apply to the availability of these data, which were used under license for the current study, and so are not publicly available. For reusing these data, an application must be made directly to CPRD.

**ORCID iDs**
Elizabeth Ford http://orcid.org/0000-0001-5613-8509
Jackie A Cassell http://orcid.org/0000-0003-0777-0385

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
