## [Reviewer comments · BMJ Open]

ARTICLE DETAILS

TITLE (PROVISIONAL)	Automated detection of patients with dementia whose symptoms have been identified in primary care but have no formal diagnosis: a retrospective case-control study using electronic primary care records
AUTHORS	Ford, Elizabeth; Sheppard, Joanne; Oliver, Seb; Rooney, Philip; Banerjee, Sube; Cassell, Jackie

VERSION 1 – REVIEW

REVIEWER	Charlotte Warren-Gash London School of Hygiene & Tropical Medicine
REVIEW RETURNED	20-Jul-2020

GENERAL COMMENTS	Summary In this matched case control study, 46,713 patients aged 65 years or over with a dementia diagnosis in their primary care record were matched 1:1 on age, sex and practice to patients without a dementia diagnosis. Three machine learning approaches were used to investigate whether codes or keyword concepts to indicate symptoms or interventions for dementia could identify patients with dementia but without a diagnostic code in their EHR. These approaches gave AUCs of 0.87-0.90 based on coded variables alone, rising to 0.90-0.94 with the addition of keywords. Originality/ importance The proportion of people with dementia who have been formally diagnosed has risen over time from around 50% to approximately 67% with various UK Government initiatives to incentivise dementia recording in primary care. Nevertheless, one third remain without a formal diagnosis. While diagnostic rates of 100% are not necessarily desirable, given the potential stigma and psychological impact of a dementia diagnosis and the lack of effective interventions, improving diagnostic coding could be useful for surveillance and delivery of care as well as for dementia research using EHRs. Comments 1. In the introduction, there was some information about reluctance to diagnose dementia among GPs, which was helpful to highlight the debate about 'early' versus 'timely' diagnosis(1). Nevertheless, the assumption was that, among the undiagnosed, there are two groups of patients – those whose memory loss has not been identified by the GP, and those who are 'known but unlabelled' – with this study aiming to identify the latter group. There was an implicit assumption that all in this group should be labelled. However, it should be acknowledged that this group might be heterogeneous, e.g. comprising both people with no diagnostic
--

	label due to oversight or administrative error, along with those for whom the GP has made a conscious decision not to record a dementia diagnosis. Further discussion of the ethics and desirability of an algorithm labelling this final group would be useful. 2. Dementia diagnoses recorded in secondary care or on death certificates are often used in electronic health record studies to improve the sensitivity of recording. Given that CPRD has links to Hospital Episode Statistics and death certificate data from the Office for National Statistics, it was not clear why such links were not used in this study. 3. The study time period was 2000-2012, which is before many of the recent incentives to improve dementia diagnoses in primary care. The time period was presumably selected as the last time when free text and clinic letters were accessible to researchers through CPRD. Nevertheless, given recent secular trends in dementia recording, it was unclear whether results from this time period would apply in 2020. It would be helpful to see both a clearer rationale for selecting this time period and a discussion of any limitations of the approach. 4. On p7 of the methods it was not clear whether patients needed to have three or five years of data prior to the index date. 5. The results were generally well-presented. However, in figure 2, it was difficult to see the logistic regression ROC. Could this be made clearer? 6. Could the annex of Alzheimer's prescription codes also include a high level description for each product code? 7. In the discussion, it would be good to reference the recent study by Aldus et al(2) about the characteristics of patients with undiagnosed dementia in primary care. 8. Recommendations for further validation research were nicely laid out in the discussion and conclusion. Based on this study what are your recommendations for researchers using EHR data for dementia studies? E.g. should they undertake sensitivity analyses using additional concepts to capture dementia diagnoses? References 1. Brayne C, Kelly S. Against the stream: early diagnosis of dementia, is it so desirable? BJPsych Bull. 2019 Jun;43(3):123–5. 2. Aldus CF, Arthur A, Dennington-Price A, Millac P, Richmond P, Denning T, et al. Undiagnosed dementia in primary care: a record linkage study. Health Serv Deliv Res. 2020 Apr;8(20):1–108.
--	--

REVIEWER	Anna Ponjoan IDIAP Jordi Gol (Catalonia)
REVIEW RETURNED	31-Jul-2020

GENERAL COMMENTS	Many thanks for the opportunity to review this interesting and well-written manuscript. The rationale for identifying dementia cases amongst no-diagnosed patients using electronic health records from primary care is very relevant. Besides, the study was based on a large sample size of more than 93.000 persons. I commend
---

the authors for their efforts. However, there are a number of issues that need to be addressed before we will consider the paper publication.

Introduction

- Paragraph 3 is confusing because the described initiatives were subsequent to the data collected in this study. Please specify the time when these initiatives were implemented.

- Paragraph 4: "...a true prevalence of 6.6% ..." The term "true" is unclear, please remove the term "true" or change it for a more accurate word.

- Paragraph 5: please remove "that" in the sentence "GPs have reported that that diagnosing dementia..."

- Could you please provide further details about the English health care system in relation to dementia? Do GPs make the diagnosis themselves without referring? The prescription of anti-dementia drugs is made by the GP or a specialist? There is an expert group that reviews the prescription of anti-dementia drugs?

Institutionalized patients are in contact with their GP or primary care services? Which patients are screened for dementia?

Methods

- It is not clear to me why you used data from 2000-2012, instead of recent data (up to 2019 for example). Please could you clarify this point in the text?

- Could you please further explain the CPRD GOLD database?

What are the differences between CPRD and CPRD GOLD? If you used data uniquely from CPRD GOLD I would suggest explaining it in the "Data source" section.

- Study population: "Control patients were randomly sampled from patients who had no dementia codes anywhere in their record..." dementia codes refer solely to a diagnosis of dementia or any code related to dementia -such as pharmacological treatment?

Please specify which dementia codes were considered and how this issue could enhance selection bias. When should controls not have a dementia record, at "index date", at 31/12/2012?

- Study population: Could you please justify why the matching was performed 1:1 instead of 1 dementia diagnosed patient versus several persons without dementia code? Since the under-register of dementia diagnoses is quite important, it is plausible that some true dementia cases without a diagnostic code registered in their electronic health records could be considered as control.

- Study population: I do not properly understand if controls should be alive on December, 31th 2012, or death could occur at any time after the index day. Please clarify this point.

- Study population: "The entire coded patient record was extracted..." entire refers to all codes even if they were not related to dementia? or it refers to a specific period of time? Which one?

- The subsection "Refining participants" is confusing to me. Some details described in the "refining participants" section are incongruent with the explanations provided in the "Study population". For example, patients were required to have records available for at least three years according to the "Study Population", but at least five years according to the "refining participant" section. Please clarify this point. I would strongly recommend to remove the "refining participant" section and to provide all the information in the "Study population".

- Feature selection: in the appendix, you provided the Alzheimer's prescription Product Codes, which are specific for CPRD. To reach a broader audience, please provide a brief and generic description of which pharmacological treatments were included.

- Analyses: Please provide more details in the method section on how you have calculated PPV: which was the numerator? and the denominator?
- Analyses: could you please specify the number of features included in each model?
- Analyses: please describe the characteristics of each algorithm (random forest, logistic regression, and naïve Bayes) and the main differences between them. This explanation could be very helpful for the audience who are not familiarized with these techniques.

Results

- I'm wondering if you could describe the new dementia cases identified by means of the algorithms you performed. For example, it could be of interest to describe if they were more prone to live in care homes, if they have higher levels of morbidity or less life expectancy, or if they belong to certain socioeconomic status and are not assisted by public health system but by private clinics.
- Time, when symptoms were registered, could be of really helpful to understand the under-register of the "known but unlabelled patients". Please describe how long the symptoms were recorded before "index date". Furthermore, provide information about how long these patients had registers after the "index date"? this information could be helpful to understand if these patients died, were institutionalized or left CPRD (that is moved to another health provider) earlier than dementia diagnosed patients.

Discussion

- As far as I understood, dementia treated but not diagnosed patients were considered as controls. In other primary care databases patients treated with anti-dementia drugs without having a dementia diagnosis were approximately 10% of dementia cases (see Ponjoan et al. 2019; doi: 10.2147/CLEP.S186590). Please discuss the role of anti-dementia drugs prescription as a possible indicator of dementia cases and compare your findings with the ones observed in other large primary care databases.
- In the discussion, I miss an explanation about the contribution of your results to the dementia research based on real-world data. Please provide guidelines or recommendations for researchers who are interested in identifying dementia cases in large clinical databases. Which variables/features are the most important to assess under-register? Which technique (logistic regression, naïve Bayes, or random forest) would you recommend to conduct in future studies?
- I understood that the performance of an MMSE test was considered in the study but not the score obtained. Please discuss how this lacking information could affect your results.
- In the discussion, I miss a subsection or a paragraph about "clinical implications". Please explain the utility of the models performed, how these models will be used in the primary care services? Furthermore, please discuss how these clinical implications will be influenced by the estimated PPV: how a PPV of 0.4-0.5 might affect the aim to "identify known but unlabelled patients with dementia in primary care"?
- In the UK the national health institute has been making a great effort to reduce under-register of dementia diagnoses amongst persons aged 65 years or older. From 2013 primary care incentive schemes were introduced to increase dementia diagnosis rates. According to Mason et al. 2017 (DOI: 10.1002/gps.4897) these policies were effective to reduce under-register of dementia diagnoses in primary care. Since your dataset includes data until 2012, I understand that your study did not include the effect of

	these policies. Do you think that the algorithm you developed could be accurate when applied in data obtained after 2013? How these policies might affect your results and clinical implications of your study?
--	---

VERSION 1 – AUTHOR RESPONSE

Reviewer 1		
In the introduction, there was some information about reluctance to diagnose dementia among GPs, which was helpful to highlight the debate about ‘early’ versus ‘timely’ diagnosis(1 Brayne C, Kelly S. Against the stream: early diagnosis of dementia, is it so desirable? BJPsych Bull. 2019 Jun;43(3):123–5). Nevertheless, the assumption was that, among the undiagnosed, there are two groups of patients – those whose memory loss has not been identified by the GP, and those who are ‘known but unlabelled’ – with this study aiming to identify the latter group. There was an implicit assumption that all in this group should be labelled. However, it should be acknowledged that this group might be heterogeneous, e.g. comprising both people with no diagnostic label due to oversight or administrative error, along with those for whom the GP has made a conscious decision not to record a dementia diagnosis. Further discussion of the ethics and desirability of an algorithm labelling this final group would be useful.	Thank you for these insightful comments. We have expanded the introduction to include reference to the Brayne commentary and have added a sentence to indicate that the GP would need to evaluate the cases identified by the algorithm to determine if labelling them were desirable. We have added further discussion on the ethics of the approach, and the advisability of conducting research with GPs on how they might use the output of the algorithm to the discussion. We especially appreciated the suggestion to explore the ethics of early detection in this this paper and have cited several authors who have written on this topic.	5 and 14-15
Dementia diagnoses recorded in secondary care or on death certificates are often used in electronic health record studies to improve the sensitivity of recording. Given that CPRD has links to Hospital Episode Statistics and death certificate data from the Office for National Statistics, it was not clear why such links were not used in this study.	We did not have access to linked data for this study (due to financial constraints when accessing the database) but will make use of it in follow up studies. We have mentioned this in the discussion.	16
The study time period was 2000-2012, which is before many of the recent incentives to improve dementia diagnoses in primary care. The time period was presumably selected as the last time when free text and clinic letters were accessible to researchers through CPRD. Nevertheless, given recent secular trends in dementia recording, it was unclear whether results from this time period would apply in 2020. It would be helpful to see both a clearer rationale for	We have added a sentence in the methods, in the description of CPRD free text, describing why the cessation of collection of free text limited our study period to the end of 2013. We have also drawn attention to this limitation at the beginning of the study limitations section.	7 and 16

selecting this time period and a discussion of any limitations of the approach.		
On p7 of the methods it was not clear whether patients needed to have three or five years of data prior to the index date.	For the extraction from CPRD, we stipulated each patient had to have 3 years of records before index date. We then refined the extracted sample in house, to use on those patients with at least 5 years records prior to index date. We have made this clearer in the text.	7-8
The results were generally well-presented. However, in figure 2, it was difficult to see the logistic regression ROC. Could this be made clearer?	The figures supplied were high resolution and clear, so we think this may be a PDF conversion problem in the submission system.	No change
Could the annex of Alzheimer's prescription codes also include a high level description for each product code?	We have completed a table of full details of product code descriptions in Appendix 2.	Appendix 2
In the discussion, it would be good to reference the recent study by Aldus et al(2) about the characteristics of patients with undiagnosed dementia in primary care	Thanks for drawing our attention to this very interesting new paper. We have cited the paper and its findings in the discussion	14-15
Recommendations for further validation research were nicely laid out in the discussion and conclusion. Based on this study what are your recommendations for researchers using EHR data for dementia studies? E.g. should they undertake sensitivity analyses using additional concepts to capture dementia diagnoses?	Making suggestions for improving EHR research quality is a good idea and we have developed these section in the discussion.	17-18
Reviewer: 2		
Paragraph 3 is confusing because the described initiatives were subsequent to the data collected in this study. Please specify the time when these initiatives were implemented.	We have added dates and extra references into the text.	4
Paragraph 4: "...a true prevalence of 6.6% ..." The term "true" is unclear, please remove the term "true" or change it for a more accurate word.	We have removed the word "true".	5
Paragraph 5: please remove "that" in the sentence "GPs have reported that that diagnosing dementia...".	We have removed the extra word "that"	5
Could you please provide further details about the English health care system in relation to dementia? Do GPs make the diagnosis	We have added a description of the UK healthcare pathway to dementia diagnosis and follow up care to the	4

themselves without referring? The prescription of anti-dementia drugs is made by the GP or a specialist? There is an expert group that reviews the prescription of anti-dementia drugs? Institutionalized patients are in contact with their GP or primary care services? Which patients are screened for dementia?	introduction for the benefit of international readers.	
It is not clear to me why you used data from 2000-2012, instead of recent data (up to 2019 for example). Please could you clarify this point in the text?	We used this duration of data because of using CPRD free text in the study. CPRD stopped collecting free text mid 2013, so we took data up to the last full year when free text was available. We have explained this reasoning in the methods section.	7
Could you please further explain the CPRD GOLD database? What are the differences between CPRD and CPRD GOLD? If you used data uniquely from CPRD GOLD I would suggest explaining it in the "Data source" section.	We have added an explanation of the two CPRD databases in the methods section and clarified that we used only CPRD GOLD.	7
Study population: "Control patients were randomly sampled from patients who had no dementia codes anywhere in their record..." dementia codes refer solely to a diagnosis of dementia or any code related to dementia -such as pharmacological treatment? Please specify which dementia codes were considered and how this issue could enhance selection bias. When should controls not have a dementia record, at "index date", at 31/12/2012?	Thank you for requesting this clarification. The controls had no dementia code at any date in their record from their moment of registration to the date they left the practice or the date of last database build which included free text (which was November 2013).	8
Study population: Could you please justify why the matching was performed 1:1 instead of 1 dementia diagnosed patient versus several persons without dementia code? Since the under-register of dementia diagnoses is quite important, it is plausible that some true dementia cases without a diagnostic code registered in their electronic health records could be considered as control.	At the time of data extraction we were constricted to 100,000 patients for financial reasons (extracting more than this number cost more money). We extracted all eligible dementia patients (47~k) and so it was only possible to match one-to-one. In fact, not every dementia case could be found a match due to our tight matching criteria. We acknowledge this as a limitation and have mentioned it in the discussion.	16
Study population: I do not properly understand if controls should be alive on December, 31th 2012, or death could occur at any time after the index day. Please clarify this point.	Death could occur any time after the index date. We have added an explanation about this to the methods.	8

Study population: “The entire coded patient record was extracted...” entire refers to all codes even if they were not related to dementia? or it refers to a specific period of time? Which one?	This refers to the entire patient record by date and by all codes. From the date of registration of the patient to the last date of data extraction or the date of patient death or leaving the practice, we took all codes for each study patient. We have explained this further.	8
The subsection “Refining participants” is confusing to me. Some details described in the “refining participants” section are incongruent with the explanations provided in the “Study population”. For example, patients were required to have records available for at least three years according to the “Study Population”, but at least five years according to the “refining participant” section. Please clarify this point. I would strongly recommend to remove the “refining participant” section and to provide all the information in the “Study population”.	There were two stages: First we sent an extraction specification to CPRD to extract a study population on our behalf and deliver us the data. Second we made some new decisions to refine the sample based on what we decided were our study needs. We cut out patients who had less than 5 years data recorded at this stage. We hope this is made clearer now in the text.	7-8 8
Feature selection: in the appendix, you provided the Alzheimer’s prescription Product Codes, which are specific for CPRD. To reach a broader audience, please provide a brief and generic description of which pharmacological treatments were included.	We have now provided a table with a product description for each code	Appendix 2
Analyses: Please provide more details in the method section on how you have calculated PPV: which was de numerator? and the denominator?	We have added more detail about this method.	10
Analyses: could you please specify the number of features included in each model?	We have included this more explicitly, that our 8 coded concepts were included in all models, with the additional 9 keyword concepts added to the keyword models.	9
Analyses: please describe the characteristics of each algorithm (random forest, logistic regression, and naïve Bayes) and the main differences between them. This explanation could be very helpful for the audience who are not familiarized with these techniques.	Descriptions of each method has been added to the analysis section together with some references	9-10
As far as I understood, dementia treated but not diagnosed patients were considered as controls. In other primary care databases patients treated with anti-dementia drugs without having a dementia diagnosis were approximately 10% of dementia cases (see Ponjoan et al. 2019; doi: 10.2147/CLEP.S186590). Please discuss the	Thank you for this observation. We decided not to clean the controls with dementia prescriptions out of the sample, because this was one of our predictive features, and we felt if we had already removed this from our control group, it would inflate our	16

role of anti-dementia drugs prescription as a possible indicator of dementia cases and compare your findings with the ones observed in other large primary care databases.	model performance unfairly. We have therefore taken your advice to discuss this as a limitation in the discussion, to reference the Ponjoan paper, and suggest removing controls with dementia medication as a refinement in future research.	
In the discussion, I miss an explanation about the contribution of your results to the dementia research based on real-world data. Please provide guidelines or recommendations for researchers who are interested in identifying dementia cases in large clinical databases. Which variables/features are the most important to assess under-register? Which technique (logistic regression, naïve Bayes, or random forest) would you recommend to conduct in future studies?	We have added some suggestions on page 17	17
I understood that the performance of an MMSE test was considered in the study but not the score obtained. Please discuss how this lacking information could affect your results.	We have added this as a limitation and an avenue for future exploration. We have also added an explanation in the methods as to why we did not use MMSE score.	9, 16-17
In the discussion, I miss a subsection or a paragraph about “clinical implications”. Please explain the utility of the models performed, how these models will be used in the primary care services? Furthermore, please discuss how these clinical implications will be influenced by the estimated PPV: how a PPV of 0’4-0’5 might affect the aim to “identify known but unlabelled patients with dementia in primary care”?	We have added text on exploring implications of the model with stakeholders, and the ethical implications of the approach to the new discussion section on page 15. We have mentioned the low PPV and its implications in this section also.	15
In the UK the national health institute has been making a great effort to reduce under-register of dementia diagnoses amongst persons aged 65 years or older. From 2013 primary care incentive schemes were introduced to increase dementia diagnosis rates. According to Mason et al. 2017 (DOI: 10.1002/gps.4897) these policies were effective to reduce under-register of dementia diagnoses in primary care. Since your dataset includes data until 2012, I understand that your study did not include the effect of these policies. Do you think that the algorithm you developed could be accurate when applied in data obtained after 2013? How these policies might affect your results and clinical implications of your study?	We have added a new sentence highlighting the possible change in clinical practice and thus recording of data given new initiatives, and the need to replicate this study in more recent data (which we intend to do). However, free text data would no longer be available in a newer study.	16-17

Formatting		
1. Required Supplementary format:		
- Please re-upload your Supplementary files in PDF format		PDFs
2. Appendix 4 citation missing:		
- The in text citation for "Appendix 4" is missing in the main text of your main document file. Please amend accordingly	This is a reporting checklist so there is not really anywhere to cite it but we have included it in the study design section of the methods.	6

VERSION 2 – REVIEW

REVIEWER	Charlotte Warren-Gash London School of Hygiene & Tropical Medicine
REVIEW RETURNED	30-Sep-2020

GENERAL COMMENTS	I am satisfied that my original comments have been adequately addressed.
--

REVIEWER	Anna Ponjoan ISV-Girona group Institu d'Investigació en Atenció Primària Jordi Gol (IDIAPJGol)
REVIEW RETURNED	16-Oct-2020

GENERAL COMMENTS	Many thanks for the opportunity to review this improved version of the manuscript. I'm still having some doubts about not considering as dementia cases those persons treated with anti-dementia drugs. In my opinion, considering only diagnosed patients as dementia cases and including the feature "Alzheimer's Medication" artificially increases the performance of the model. I'm pretty sure that having a dementia code for prescription of anti-dementia drugs indicates that this person is a dementia case, even more, likely than those persons having diagnosis code for dementia. The authors added this sentence to the reviewed manuscript: "We also chose not to clean up our control group for evidence of dementia medication prescribing as has been done in other studies [56] because we felt that doing so would have artificially inflated the performance of our models". Under my point of view, this point is very important and deserves further attention. Could you please provide sensitivity analyses which included both diagnosed and treated patients were considered dementia cases (and excluding the "Alzheimer's Medication" from the analyses) to confirm your feelings about an expected inflation of the performance? Sensitivity analyses can be added as supplemental material. Several variables included in the models (Cognitive decline, cognitive screening test, dementia annual review, memory loss codes, MMSE, referral to memory assessment, as well as several keywords) could be indicative of mild cognitive impairment. I'm wondering that a certain amount of cases identified by the models
---

	would not be dementia patients but, for example, persons affected by mild cognitive impairment. Thus, I would send a cautious message rather than “These findings also provide a potential way for EHR researchers to extend their dementia case identification when using primary care data for dementia research and potentially to improve research quality.” Please note that the PPV is quite low. The only way to ensure a good identification of dementia cases when using EHR is to conduct a validation study. This need to be further discussed and clearly satated in the limitations.
--	--

VERSION 2 – AUTHOR RESPONSE

Many thanks once again for engaging such interested reviewers for our manuscript and for inviting us to submit a revision. We thank Reviewer 2 for continuing to suggest improvements to our report.

We have responded to her two new suggestions fully and detail the changes we have made below.

Reviewer 2 Comment 1: I’m still having some doubts about not considering as dementia cases those persons treated with anti-dementia drugs. In my opinion, considering only diagnosed patients as dementia cases and including the feature “Alzheimer’s Medication” artificially increases the performance of the model. I’m pretty sure that having a dementia code for prescription of anti-dementia drugs indicates that this person is a dementia case, even more, likely than those persons having diagnosis code for dementia. The authors added this sentence to the reviewed manuscript: “We also chose not to clean up our control group for evidence of dementia medication prescribing as has been done in other studies [56] because we felt that doing so would have artificially inflated the performance of our models”. Under my point of view, this point is very important and deserves further attention. Could you please provide sensitivity analyses which included both diagnosed and treated patients were considered dementia cases (and excluding the “Alzheimer’s Medication” from the analyses) to confirm your feelings about an expected inflation of the performance? Sensitivity analyses can be added as supplemental material.

Our Response: We ran sensitivity analyses as suggested. We removed all controls who had Alzheimer’s medication and additionally, all controls who had dementia annual review codes, as we felt this was also indicative of a prior diagnosis. This totalled N= 300 control patients. We also re-ran all the models without these two code lists as predictors. Interestingly, the codes-only model performed substantially worse (AUC 0.83), but specificity improved (0.95). The codes and keywords models largely maintained their performance (AUC 0.93). We have described the sensitivity analysis in the methods (page 10-11) given an overview of results in the results section (page 13), discussed these in context (page 15) and presented the full set of results (feature weights, model accuracy and ROC curves) in the new supplementary material (Appendix 4).

Reviewer 2 Comment 2: Several variables included in the models (Cognitive decline, cognitive screening test, dementia annual review, memory loss codes, MMSE, referral to memory assessment, as well as several keywords) could be indicative of mild cognitive impairment. I’m wondering that a

certain amount of cases identified by the models would not be dementia patients but, for example, persons affected by mild cognitive impairment. Thus, I would send a cautious message rather than "These findings also provide a potential way for EHR researchers to extend their dementia case identification when using primary care data for dementia research and potentially to improve research quality." Please note that the PPV is quite low. The only way to ensure a good identification of dementia cases when using EHR is to conduct a validation study. This need to be further discussed and clearly stated in the limitations.

Our response: This is a good point, thank you. We have introduced caveats in two areas of the discussion; one in the limitations section, where we highlight that patients detected by our model might have mild cognitive impairment or another non-dementia memory complaint, and that a real life evaluation would help us understand more about potential false positives (page 18). In the implications section we have suggested that any new case identification methods using these combinations of predictors should be subject to validation studies (page 18-19).

In summary we feel these changes have resulted in additional improvements to our manuscript and we hope that you now find our manuscript is ready for acceptance in BMJ Open.

We look forward to hearing from you in due course.